# How Leaf Vein and Stomata Traits Are Related with Photosynthetic Efficiency in Falanghina Grapevine in Different Pedoclimatic Conditions

**DOI:** 10.3390/plants11111507

**Published:** 2022-06-04

**Authors:** Nicola Damiano, Carmen Arena, Antonello Bonfante, Rosanna Caputo, Arturo Erbaggio, Chiara Cirillo, Veronica De Micco

**Affiliations:** 1Department of Agricultural Sciences, University of Naples Federico II, Via Università 100, 80055 Portici, Italy; nicola.damiano@unina.it (N.D.); rosanna.caputo@unina.it (R.C.); 2Department of Biology, University of Naples Federico II, Via Cinthia 21-26, 80126 Napoli, Italy; c.arena@unina.it; 3Institute for Mediterranean Agricultural and Forest Systems, ISAFOM, National Research Council of Italy (CNR), P.le Enrico Fermi 1, 80055 Portici, Italy; antonello.bonfante@cnr.it (A.B.); arturo.erbaggio@isafom.cnr.it (A.E.)

**Keywords:** climate changes, leaf traits, photosynthesis, vein and stomata traits, *Vitis vinifera*

## Abstract

The increase in severe drought events due to climate change in the areas traditionally suitable for viticulture is enhancing the need to understand how grapevines regulate their photosynthetic metabolism in order to forecast specific cultivar adaptive responses to the changing environment. This study aims at evaluating the association between leaf anatomical traits and eco-physiological adjustments of the ‘Falanghina’ grapevine under different microclimatic conditions at four sites in southern Italy. Sites were characterized by different pedoclimatic conditions but, as much as possible, were similar for plant material and cultivation management. Microscopy analyses on leaves were performed to quantify stomata and vein traits, while eco-physiological analyses were conducted on vines to assess plant physiological adaptation capability. At the two sites with relatively low moisture, photosynthetic rate, stomatal conductance, photosystem electron transfer rate, and quantum yield of PSII, linear electron transport was lower compared to the other two sites. Stomata size was higher at the site characterized by the highest precipitation. However, stomatal density and most vein traits tended to be relatively stable among sites. The number of free vein endings per unit leaf area was lower in the two vineyards with low precipitation. We suggest that site-specific stomata and vein traits modulation in Falanghina grapevine are an acclimation strategy that may influence photosynthetic performance. Overall in-depth knowledge of the structure/function relations in Falanghina vines might be useful to evaluate the plasticity of this cultivar towards site-specific management of vineyards in the direction of precision viticulture.

## 1. Introduction

Nowadays, climate change is challenging agriculture since it can drastically modify plant growth, with possible negative effects, especially in arid and semi-arid regions of Europe. In the Mediterranean area, climate models often show irregularities in precipitation patterns and significantly rising temperatures leading to the increase in frequency, severity, and duration of drought events [1,2,3]. The interest in understanding how Mediterranean crops face drought is currently increasing due to the severe limitations expected in plant growth and productive yield in the future [4,5].

Grapevine (*Vitis vinifera* L. subsp. *vinifera*) is a high-income crop, rain-fed cultivated in many Mediterranean areas, according to the specific requirements of quality and origin labels. The productivity of the ‘Falanghina’ cultivar, which is important in southern Italy, is expected to be severely impacted by environmental changes. The typical temperature and precipitation regimes during summer lead to a decrease in leaf area and photosynthesis in the water-stressed vines, ultimately causing physiological and metabolic disorders with negative effects on the overall plant functioning, including nutrient uptake, fruit set, and berry ripening [6,7]. Under water stress conditions, many adaptation mechanisms can occur mainly related to an increase in water holding capacity, decrease in water losses, and mechanical reinforcement to prevent any tissue damage [8]. For instance, one of the first plant responses to water deficit is a decrease in the investment in leaves compared to other organs due to a change in carbon partitioning favoring the flow of assimilates towards the root [9]. In the current climate change scenario, the occurrence of osmotic stress due to soil and water salinization can also affect plants' gas exchange and lead to leaf anatomy adjustments similar to those observed in response to water stress [10]. All these morpho-physiological alterations affect both yield and berry composition (e.g., soluble solids, organic acids, polyphenols), often associated with decreasing must quality [11]. In many areas of southern Italy, grapevines are subjected to water stress when high evapotranspiration is accompanied by low precipitation [12], and it has been emphasized that strategies engaged by plants to mitigate the environmental stress must be based on a deep knowledge of plant plasticity in terms of structure/functions relationships [13,14].

An important question in many crops, including grapevine, is how plants efficiently produce leaves capable of supplying enough water to balance the transpiration losses. In the regulation of this mechanism, it is critical that plants have a satisfactory equilibrium between the stomata density/size, which controls maximum stomatal conductance and the transpiration rate [15], and leaf vein density, which regulates water supply throughout the leaf tissues [16,17]. Generally, the balance between the investment in vapor and liquid conductance in the leaf is well conserved in plant groups along evolutionary trends [18].

In the open field, under saturating light conditions, the most efficient combination of stomata and vein investment is reached when the soil water supply is enough to maintain stomata fully open [19]. However, if the vascular system is not sufficiently developed to support the maximum evaporative capacity of the leaves, when the water supply is limited, stomata closure occurs to maintain leaf water status [20]. The harmonization between stomata and vein traits is a delicate question, as an excessive stomatal density may determine high costs for the construction and regulation of guard cells that are not necessary for greater photosynthetic yield when stomata are closed. Similarly, the venation excess may not be efficient when photosynthesis declines (due to increased leaf volume occupied by the vascular system to the detriment of photosynthetic parenchyma) and the cost of synthesizing lignin increases [21]. The coordination between water transport and stomatal systems allows leaves to maintain an efficient balance between water use and carbon gain while accommodating the different rates of photosynthesis and transpiration experienced under high and low irradiance [22,23].

Based on the stomatal regulation, grapevine cultivars have been classified as isohydric or anisohydric, with isohydric vines being able to promptly regulate stomatal responses to maintain constant water potential, and anisohydric vines close their stomata only when water potential is very low [24]. Efforts to relate such behavior with leaf and stem anatomical traits are reported for a few cultivars [24,25,26]. The isohydric behavior is associated with higher stomata frequency and larger vessels in the stem, thus to a higher hydraulic conductance (corresponding to higher vulnerability to embolism) compared to anisohydric models, whose anatomical traits allow delaying stomatal closure and reaching lower water potentials without xylem cavitation [24]. Therefore, anisohydric behavior would allow a more efficient carbon fixation under short-term mild stress [27]. However, the classification of grapevine cultivars as isohydric or anisohydric is still under debate, and it seems that the same cultivar can show either behavior depending on environmental conditions, such as the severity and duration of the stress event [26,28].

Apart from the stomatal behavior, the photosynthetic rate in plants is also influenced by the transfer resistance for CO_2_ diffusion throughout the mesophyll, which contains two main components. The first regards the pathway of CO_2_ diffusion from the sub-stomatal cavity to the outer surface of mesophyll cells and is related to the 3D pattern of intercellular spaces; the second involves the path to reach the carboxylation sites in the chloroplasts and is influenced by the permeability of cell walls of the photosynthetic cells [26]. Both components have been found to contribute to a higher photosynthetic rate in *V. vinifera* ‘Ribier’ compared to *Vitis labrusca* ‘Isabella’, while differences in photosynthesis among *V. vinifera* ‘Athiri’, ‘Asyrtiko’, and ‘Syrah’ have been mainly ascribed to the resistance across cell walls [29,30]. However, more recently, the variability in mesophyll conductance in seven grapevine cultivars has been shown to be independent of mesophyll anatomical parameters [31].

In the last decade, grapevine morpho-anatomy has been claimed as an understudied topic with a possible important impact on functional responses of vines to environmental stress factors. Leaf epidermal, stomata, and mesophyll traits have been studied in relation to physiological traits only in a few cultivars. Therefore, there is an increasing need to expand knowledge of vine structural and eco-physiological plasticity to finely design precision viticulture strategies for the implementation of irrigation management plans [28,32]. To the best of our knowledge, the strict relations between stomata traits and leaf venation have not been analyzed yet to infer their role in the physiological adjustment of vines growing under limiting environmental conditions.

In this framework, the aim of this study is to better assess the coordination between leaf vein and stomata traits and eco-physiological parameters in *Vitis vinifera* L. subsp. v*inifera* ‘Falanghina’ grown at four sites in southern Italy. More specifically, we aimed to evaluate how anatomical and eco-physiological parameters are coordinated under different pedoclimatic conditions. We focused on the veraison phenological phase, which corresponds to maximum vegetative growth when water availability is limited in semi-arid Mediterranean environments.

## 2. Results

### 2.1. Environmental Data Characterization

The weather information collected from the Guardia Sanframondi station during the phenological phase of veraison showed that temperature was similar in 2019 and 2020, with a July average temperature of 26.9 °C (SD ± 2.6) and 26.0 °C (SD ± 2.0), with a maximum temperature of 34.0 °C and 34.3 °C, and with a minimum of 20.4 °C and 19.1 °C, respectively, in 2019 and 2020. July 2020 tended to have less rainfall (14 mm) and lower ET_0_ (120 mm) than 2019 (29 mm rainfall, 176 mm ET_0_). In 2020, it was possible to measure the cumulative precipitation separately in each experimental vineyard: 32mm SL, 15mm CA, 18 mm GR, and 9 mm AC. Supplemental irrigation was provided at the AC site; therefore, the vineyards could be grouped as SL and AC receiving relatively abundant moisture, whereas CA and GR had relatively limited moisture.

Soil water content (SWC) at 15 and 30 cm depth was higher in CA compared to the other sites. At 75 cm depth, GR had the highest SWC, followed by CA. SWC values of SL and GR tended to increase with increasing depth (Appendix A). The soil temperature decreased with increasing depth at SL, GR, and AC (Appendix A).

### 2.2. Growth and Production Parameters

Growth and production parameters (total shoot leaf area, single leaf area, shoot basal diameter, bunch weight, and number) are reported in Table 1. The main effect of field (F) was significant for all analyzed parameters; the year (Y) as the main factor showed a significant effect on all factors but bunch weight (Table 1).

In particular, the average total shoot leaf area was significantly higher in SL and GR compared to CA and AC. Single leaf area was significantly higher in SL than in AC and GR, which, in turn, showed significantly higher values than CA. Shoot basal diameter was higher in GR than AC which resulted in being significantly higher than CA, while SL showed intermediate values between GR and AC. For average bunch weight, SL was higher than AC, which was significantly higher than CA. The lowest yield was found in GR. For bunch number, AC was higher than SL and GR, which in turn showed significantly higher values than CA. All parameters except bunch weights were higher in 2019 than in 2020.

The interaction between field and year (F x Y) was significant only in the case of bunch weight and bunch number (Table 1). For bunch weight, SL always showed the highest value in 2020, while GR in 2020 was the lowest. CA showed a significant increase in bunch weight from 2019 to 2020, while the opposite significant trend occurred in GR and AC (Figure 1a). For bunch numbers in 2019, there were significant highest values for all the fields compared to 2020 (Figure 1b).

### 2.3. Gas-Exchange and Chlorophyll a Fluorescence

As regards eco-physiological parameters, the main effect of field (F) was significant for all analyzed parameters except for substomatal CO_2_ concentration (Ci) and _in_WUE. The main effect of year (Y) was significant for net photosynthetic rate (Pn), Ci, and _in_WUE; more specifically, Pn and _in_WUE showed significantly lower values in 2020 compared to 2019, while the opposite was found for Ci (Table 2). In particular, in SL leaves, Pn was significantly higher than in AC ones, which, in turn, showed higher Pn than CA and GR. The stomatal conductance (gs), transpiration rate (E), electron transport rate (ETR), quantum yield of PSII electron transport rate (ΦPSII), and maximum PSII photochemical efficiency (Fv/Fm) showed similar values in SL and AC plants, which were also significantly higher than values measured in both CA and GR plants (Table 2).

The interaction between field and year (F x Y) was significant only for Pn (Table 2). Vines at the SL site in both years and at the AC site in 2019 showed significantly higher values than all the other conditions (Figure 2), with the lowest value recorded at GR in 2020. At all the sites, Pn decreased in 2020 compared to 2019 but significantly only in AC (Figure 2).

### 2.4. Stomata and Vein Traits

Microscopy analysis of the abaxial epidermis showed that stomata tended to be larger in leaves collected from vines at SL and CA sites, while they seemed smaller at the other sites (Figure 3). The quantification of stomata traits confirmed that there was a significant main effect of field (F) on guard cell length and width, while the main effect of year (Y) was significant for guard cell width and stomata frequency (Table 3). In particular, stomata were significantly larger at SL compared to the CA field, which in turn showed higher values than GR, whose values were significantly higher than AC (Table 3). As regards stomata frequency, only Y had a significant effect as the main factor, with values measured in 2019 leaves higher than in 2020 (Table 3).

In particular, for both stomata length and width, the fields are significantly different in the decreasing order SL, CA, GR, and AC. Considering the main effect Y, the stomata frequency was significantly higher in 2019 than in 2020, and the stomata width was significantly higher in 2020 than in 2019. The interaction (F x Y) was significant only for stomata frequency (Table 3) which was quite steady among fields in 2019. Significant differences among fields were recorded in 2020, with the lowest values in leaves collected at SL and GR sites (Figure 4).

Microscopy analysis of abaxial epidermis and quantification of vein traits evidenced that the field as the main factor had a significant effect only on Total VLA, Minor VAA, Total VAA, and FVEA (Table 4). Minor VAA was significantly higher in leaves collected at AC compared to CA and GR sites which, in turn, showed significantly higher values than SL leaves. Total VAA was significantly lower values in SL leaves compared to all the other fields (Table 4). On the contrary, SL showed significantly higher values than CA and AC leaves which in turn exhibited significantly higher values than GR.

The factor year showed a significant effect on all VAA parameters and FVEA, with values significantly lower and higher in 2020 compared to 2019 for VAA and FVEA, respectively (Table 4). The interaction (F x Y) was significant only for Total VAA and FVEA (Figure 5a,b), with values of Total VAA decreasing from 2019 to 2020 for all the fields but GR. Instead, FVEA showed increasing values from 2019 to 2020 for SL and GR, while no tendency was found in CA and AC.

## 3. Discussion

This study highlighted how Falanghina grapevine growing under different pedoclimatic conditions develops stomata and vein traits in line with different photosynthetic behavior and productivity. Anatomical and physiological traits varied among sites suggesting a different water use efficiency in the four vineyards in the two analyzed years, likely triggered by different precipitation amounts. In general, in both years, the four vineyards showed two main behaviors regarding the photosynthetic efficiency and biomass production, with SL and AC plants more performant than CA and GR ones. This agrees with a previous study in which δ^13^C values of musts were significantly higher in CA and GR vineyards, indicating they were drought-stressed compared to SL and AC ones [33]. The different growth and production performances are likely related to increased water availability due to higher precipitation levels in the case of the SL site and to the application of supplemental irrigation in the AC site, which would have compensated for the scarce amount of precipitation registered in July 2020 compared to the other study sites. The measurement of soil water content at the three soil depths also suggests that vines at SL adopted a strategy to maintain stomata open to sustain high photosynthetic rates, notwithstanding the increasing water losses through transpiration. Such a mechanism suggests that the Falanghina at the SL site would respond to water shortage conditions with slower stomata closure, also in line with the occurrence of larger stomata [24]. The risk for leaf vein embolism deriving from the delayed stomatal closure would also be prevented in vines at the SL site by virtue of the narrower veins (e.g., lower minor VAA at the same minor VLA), which are less prone to drought-induced cavitation [34,35]. Leaf embolism thresholds have been recently explored in grapevine, suggesting an important role of leaf hydraulic traits in coordination with other physiological traits to contribute to vine drought tolerance [36]. In SL vines, the occurrence of higher FVEA, compared to the vines of the other sites, also suggests a more balanced distribution of the hydraulic system across the leaf lamina, which would favor water conductivity across the mesophyll cells despite the lower VLA. High values of FVEA are associated with higher K_leaf_ (leaf hydraulic conductance) and better sugar loading in the cases when they do not correlate positively to VLA and contain phloem [37,38]. On the other hand, the Falanghina vines at the CA and GR sites were characterized by a lower net CO_2_ assimilation rate, accompanied by lower stomatal conductance and transpiration rate likely ascribed to a more efficient stomatal control due to prompt stomatal closure allowed by smaller guard cells typical of isohydric behavior. An intermediate behavior would have been assumed by AC vines which, although having anatomical traits expected for an isohydric model, were likely able to maintain stomata open due to supplemental irrigation. The higher FVEA accompanied by high total VLA, in this case, would be associated with high K_leaf_ and would support high photosynthetic efficiency. General trends to increasing VLA are reported according to growing aridity as a strategy to favor more photosynthesis during the moments of high water availability [39,40,41]. Therefore, vines at the AC site, being characterized by high VLA, high FVEA, and smallest stomata, show traits designed to benefit from supplemental irrigation leading to high photosynthetic activity and yield.

The Falanghina grapevine has been classified as a near-isohydric model, but there is evidence that cultivars classified as near-isohydric are able to change their behavior towards anisohydric status, as in Syrah [42,43]. Our findings in Falanghina suggest that this cultivar is able to acclimate eco-physiological traits by assuming different quantitative leaf stomata and vein traits under different cultivation environments. Indeed, there is evidence that vine eco-physiological behavior is dependent on the water availability in soil and the duration of water shortage [44]. In the four analyzed vineyards, stomata frequency was quite stable and within the range reported for grapevine (50–400 stomata/mm^2^) [45]. Furthermore, the observed stability of stomatal frequency across several environments agrees with the general principle by which stomata frequency is considered more an evolutionary adaptation rather than a short-term acclimation mechanism. This statement is also supported by studies reporting that limiting environmental conditions determine a more substantial effect on stomata size than on their frequency [46,47]. The environmental conditions at the early stages of growth in the Pinotage grapevine have been suggested as major determinants in modulating stomata frequency and size with implications on stomatal conductance, which, in turn, affects the whole plant water balance [48]. Indeed, this assumption has also been suggested in other species in which the environmental conditions, especially water availability, during organogenesis have been demonstrated to play a major role in the development of specific quantitative traits (e.g., stomata size and frequency, vein and xylem features) which pose the limits of physiological acclimation [49].

The leaf structure, in terms of stomata size and frequency, may have contributed to the different grapevines' capability to perform photosynthesis in the different vineyards. The higher photosynthetic rate found in SL and AC plants may be due to a higher stomatal conductance and a better PSII photochemical efficiency (i.e., elevated values of ETR, ΦPSII, Fv/Fm). The higher stomatal conductance may allow a better CO_2_ supply within substomatal chambers, thus enhancing carbon fixation [50]. Conversely, the photosynthetic activity declined significantly in CA and GR plants due not only to stomatal but also to not-stomatal limitations. Indeed, even if the stomatal closure reduced stomatal conductance (gs) and transpiration (E) in vines growing at CA and GR, substomatal CO_2_ concentration remained comparable among plants of different sites suggesting that the utilization of CO_2_ at carboxylation sites was somewhat limited [51]. The lower photochemical efficiency of CA and GR plants (i.e., reduced ΦPSII and ETR) may limit the synthesis of ATP and NADPH through the electron transport chain and could explain why the carbon fixation was lower in these plants. Under the particular environmental conditions of the sampling season, the partial closure of stomata in CA and GR plants may be interpreted as a safety strategy to avoid an excessive water loss by transpiration, thus preserving the photosynthetic apparatus from permanent damages. The low Fv/Fm values of CA and GR, which were the sites with limited precipitation and where gs was lower, compared to SL and AC, where more moisture was available, suggest that Fv/Fm may have responded to a stress condition, supporting the hypothesis that efficient avoiding strategies are needed by plants to overcome the stress [52].

Our data also indicate that the photosynthetic performance significantly depends on the year. The lower precipitation in July 2020 compared to 2019 likely caused the recorded reduction in photosynthetic levels and overall biomass production. The co-occurring decrease in Pn and _in_WUE and increase in Ci indicated that during the second season, in response to more severe stress, non-stomatal limitations occurred that contributed significantly to the reduction in carbon fixation at the carboxylation sites [53].

The overall analysis supports the idea that stomata and vein traits are likely modulated by environmental conditions during leaf development and may severely influence the physiological responses of a grapevine cultivar to short-term changes in water availability. Therefore, such traits should be considered, together with other hydraulic structural and physiological characteristics, in evaluating the drought tolerance of grapevine as also suggested by other authors, who highlighted the importance of integrating multiple traits in grapevine as already accepted for hydraulic traits in other models as forest species [36,54,55]. We suggest that only with a multi-trait approach, including the analysis of structural traits, will it be possible to have a comprehensive understanding of the single cultivar strategies adopted to cope with specific environmental constraining conditions in order to allow site-designed cultivation plans addressing the needs of precision viticulture.

## 4. Materials and Methods

### 4.1. Study Area and Vineyard Characteristics

The study area was in southern Italy in the Campania region, Guardia Sanframondi (Benevento, Figure 6), in a hilly environment characterized by a typically Mediterranean climate (cold winters and hot summers). The selected four experimental sites were placed within the vineyards of the La Guardiense farm: 1) SL-Santa Lucia, 41°14′45″ N, 14°34′16″, 194 m a.s.l.; 2) CA-Calvese, 41°14′19″ N, 14°35′11″ E, 163 m a.s.l.; 3) GR-Grottole, 41°14′21″ N, 14°34′56″ E, 158 m a.s.l.; 4) AC-Acquafredde, 41°13′44″ N, 14°35′33″ E, 84 m a.s.l. The vine cultivar studied was *Vitis vinifera* L. subsp. v*inifera* ‘Falanghina’ (Controlled designation of origin—DOC/AOC), and the four sites were selected with the criterion to identify four vineyards similar for plant material and cultivation management but different in plant water use due to pedological and microclimatic spatial variability as reported in a previous study [33].

In the four vineyards, the vines, grafted onto 157-11 Couderc rootstock, were 8–13 years old (depending on the vineyard), were spaced 1–1.25 m between plants with 2.1–2.2 m between the rows, and were trained at double Guyot. One shoot trimming was performed after the fruit set phenological phase. The vine rows of GR, CA, and AC sites were oriented E–W, while the SL site is oriented N–S. The SL, GR, and CA vineyards were cultivated in a rain-fed regime, while at AC, supplemental irrigation was applied [33].

Daily weather information (temperature, rainfall, wind, solar radiation, etc.) was collected during the experiment, in 2019 from the Guardia Sanframondi (BN) weather station (41°14′17.2″ N; 14°35′49.8″ E) of the Campania region weather network, while in 2020 from a weather station dedicated to the experiment, placed in the CA vineyard (Netsens AgriSense IoT weather station, www.netsens.it). The positioning of the Netsens weather station was determined as representative of air temperature, air humidity, wind speed, and solar radiation of all selected vineyards, considering the distance between the experimental vineyards and the landscape form (e.g., slope, aspect, elevation). Moreover, considering that, among the weather variables, rainfall is the one characterized by the highest spatial variability, a rain gauge with three FDR probes (inserted at three different soil depths, −15, −35, and −75 cm) was placed in each experimental site able to measure soil temperature and water content. The FDR probes were applied to better understand the soil water status during the growing season, given that the precipitation amount does not represent available water for plants, which depends on the combination of weather conditions and soil properties (e.g., under the same climate, two soils can have a very different water availability for the plant) [56]. The main weather information collected (e.g., temperature, solar radiation) from both weather stations in 2020 were comparable.

The soils present in the experimental sites were Mollisols, classified as Typic Calciustolls and referring to two principal soil series of the soil map of the Valle Telesina area (1:50.000) [57]: Consociazione dei suoli Pennine (SL, CA, and GR sites) and Consociazione dei suoli Taverna Starze (CA site). The soil profile was characterized by Ap and Bw horizons, and the differences between the experimental sites were principally due to the variability of the percent of stones along the soil profile and by the effect of vineyard planting, which has modified the soil horizons thickness and depth between the sites.

The relations between anatomical and functional leaf traits were analyzed by performing eco-physiological and microscopy analyses on fully expanded leaves at plant maturity over two growing seasons.

### 4.2. Biometry and Yield

The canopy of 20 vines per vineyard was characterized by performing biometrical and production measurements on 2 annual shoots per plant at the veraison phenological phase corresponding to 81 BBCH (Biologische Bundesantalt, Bundessortenamt, and Chemische Industrie). More specifically, per each shoot, the following parameters were quantified: shoot length, shoot basal diameter, number of leaves, and leaf area. At harvest (89 BBCH), the number of bunches per shoot and bunch weight were determined (weighing all bunches from the same shoots). The estimation of leaf area was performed by applying an allometric estimation model measuring the leaf lamina width in the field and applying the equations calculated based on the measurement of width and area of 20 leaves per site by means of an electronic leaf area meter (LI-3100 model, LI-COR Inc., Lincoln, NE, USA) [13,58,59].

### 4.3. Gas-Exchange and Chlorophyll a Fluorescence Emission Measurements

Leaf gas exchange and chlorophyll “a” fluorescence emission measurements were carried out on well-exposed and fully expanded leaves, characterized by similar position and exposition within the canopy per 15 plants in each site. The analyses were performed during the veraison phase of the two growing seasons, 2019 and 2020, between 10.00 and 14.00. Net CO_2_ assimilation rate (Pn), stomatal conductance (gs), substomatal CO_2_ concentration (Ci), and transpiration rate (E) were measured using an airflow rate set to 200 µmol s^−1^, at ambient CO_2_ concentration (about 400 µmol mol^−1^) and ambient temperature, with a portable infra-red gas-analyzer (LCA 4; ADC, BioScientific, Hoddesdon, UK) equipped with a broad-leaf PLC (cuvette area 6.25 cm^2^). The instantaneous water use efficiency (_in_WUE) was calculated as the ratio between Pn and E. The average VPD (vapor pressure deficit) in the leaf chamber, Tch (chamber air temperature), and RH% (relative humidity) were 5.45 kpa, 37.43 °C, and 33.33% for 2019, and 4.97 kPa, 38.76 °C, and 27.38% for 2020. Chlorophyll “a” fluorescence emission was measured using a pulse amplitude modulated portable fluorometer (Plant stress kit ADC Bioscientific Ltd., Hoddesdon, UK). Fluorescence measurements were performed on the same day as gas exchanges on the same leaves. A weak measuring of 3.4 µmol photons m^2^ s^−1^ light was used to induce the ground fluorescence signal, F_0_, on 30′ dark-adapted leaves. A saturating light pulse of 7.000 µmol photons m^2^ s^−1^ was applied to induce the maximal fluorescence level in the dark, Fm, and in the light, Fm’. The maximum PSII photochemical efficiency (Fv/Fm) was calculated as (Fm-F_0_)/Fm, and the quantum yield of PSII electron transport rate (ΦPSII) and the electron transport rate (ETR) were estimated following Genty et al. (1989) [60] and Bilger and Björkman (1990) [61]. The measurements in the light were conducted from 12:00 to 14:00 pm under environmental Photosynthetic Photon Flux Density (PPFD) ranging between 1800 and 2300 µmol photons m^2^ s^−1^.

### 4.4. Microscopy and Digital Image Analysis

At the beginning of veraison, one fully expanded leaf characterized by similar position and exposition within the canopy was collected from the same plants analyzed for eco-physiological measurements in the four vineyards. Directly in the field, the leaf samples, including the main vein, were cropped and chemically fixed in FAA (40% formaldehyde, glacial acetic acid, 50% ethanol, 5:5:90 by volume). To observe the vine leaf traits, the samples were bleached in acetone for 48 h and, when completely clear, the acetone was removed, and leaf samples were rinsed several times with distillate water. For stomatal analysis, a part of each sample was peeled off and mounted on a slide with distilled water. The remaining part of the sample was immersed in ethanol dilutions in water (30%, 50%, 70%, 100%) for 5 min each. Afterward, samples were stained in safranin for 1 min and fast Green for 15 s, rinsed in the decrescent ethanol dilutions until 100% distilled water, and mounted on a slide with distilled water [48]. The samples for stomatal analysis were observed under a BX51 epi-fluorescence microscope (Olympus, Germany) equipped with a Mercury lamp, a 330–385 nm band-pass filter, dichromatic mirror of 400 nm and above, and a barrier filter of 420 nm and above in order to detect the different auto-fluorescence emissions of stomata over the other epidermal structures [62]. For each sample, three fields were observed at 20x magnification (field area 0.237 mm^2^), and the stomatal frequency was expressed as the number of stomata per mm^2^. Images of the lamina surface from three separate regions were collected by means of a digital camera (EP50, Olympus), taking care to avoid the main veins. The digital images were analyzed with the image analysis software program CellSens 3.2 (Olympus). The size of 10 stomata per field was measured, considering both the guard cell major (pole to pole) and minor axes to calculate the area of an imaginary ellipse. The samples for vein traits analysis were mounted with distilled water on microscope slides that were observed under the BX51 light microscope (Figure 7), and for each sample, three images were collected at 5x magnification and analyzed for digital image analysis, as reported above. For leaf venation analysis, we followed Sack and Scoffoni (2013) [38] but considered the third order veins together with the higher orders vein to avoid bias in measuring because they were often looping and not easily distinguishable from higher orders.

Therefore, the analyzed parameters are as follows:minor vein length per unit area (Minor VLA) = sum of vein lengths of third or higher orders of veins divided by the difference between the area imaged and the area occupied by the second-order veins (mm/mm^2^);major vein length per unit area (Major VLA) = sum of vein lengths of second-order veins divided by the area imaged (mm/mm^2^);minor vein area per unit area (Minor VAA) = sum of vein areas of third or higher orders of veins divided by the difference between the area imaged and the area occupied by the second-order veins (mm/mm^2^);major vein area per unit area (Major VAA) = sum of vein areas of second-order veins divided by the area imaged (mm/mm^2^);total vein length per area (Total VLA) = sum of vein lengths of all order veins divided by the area imaged (mm/mm^2^);total vein area per unit area (Total VAA) = sum of vein areas of all order veins divided by the area imaged (mm^2^/mm^2^);free vein endings per unit area (FVEA) = number of vein endings divided by the area imaged (n°/mm^2^).

### 4.5. Statistical Analysis of Data

The experimental data were analyzed with the SPSS 27 statistical software (SPSS Inc., Chicago, IL, USA). The data were analyzed by two-way analysis of variance (ANOVA), considering the field (F) and year (Y) as the main factors. Whenever the interactions were significant, a one-way ANOVA was performed. Multiple comparison tests were performed with Duncan’s coefficient using *p* ≤ 0.05 as the level of probability. The Shapiro–Wilk test was performed to check for normality.

## 5. Conclusions

The research question we aimed to address was whether leaf anatomical traits related to stomata and veins in Falanghina vines develop differently in a range of field pedoclimatic conditions varying in moisture availability. We further explored the relationship between leaf anatomical traits and leaf gas exchange and photosystem attributes in these environments. At the two sites with relatively low moisture, the photosynthetic rate was lower, as was stomatal conductance, photosystem electron transfer rate, and quantum yield of PSII linear electron transport. Stomata length and width were higher at the site characterized by the highest precipitation. However, stomatal density and most vein traits tended to be relatively stable among sites. Free vein endings per unit leaf area were fewer in the two vineyards with low precipitation. We suggest that the site-specific leaf traits adjustment in Falanghina grapevine, at stomata and veins level, may represent an acclimation strategy that may influence photosynthetic performance. The findings support the hypothesis that stomata and vein traits are likely modulated by environmental, both microclimatic and pedological, conditions during leaf development and may influence the physiological responses of Falanghina grapevine to short-term changes in water availability, supporting the idea that this cultivar may behave as an isohydric or anisohydric model, as previously reported [42].

## Figures and Tables

**Figure 1 plants-11-01507-f001:**
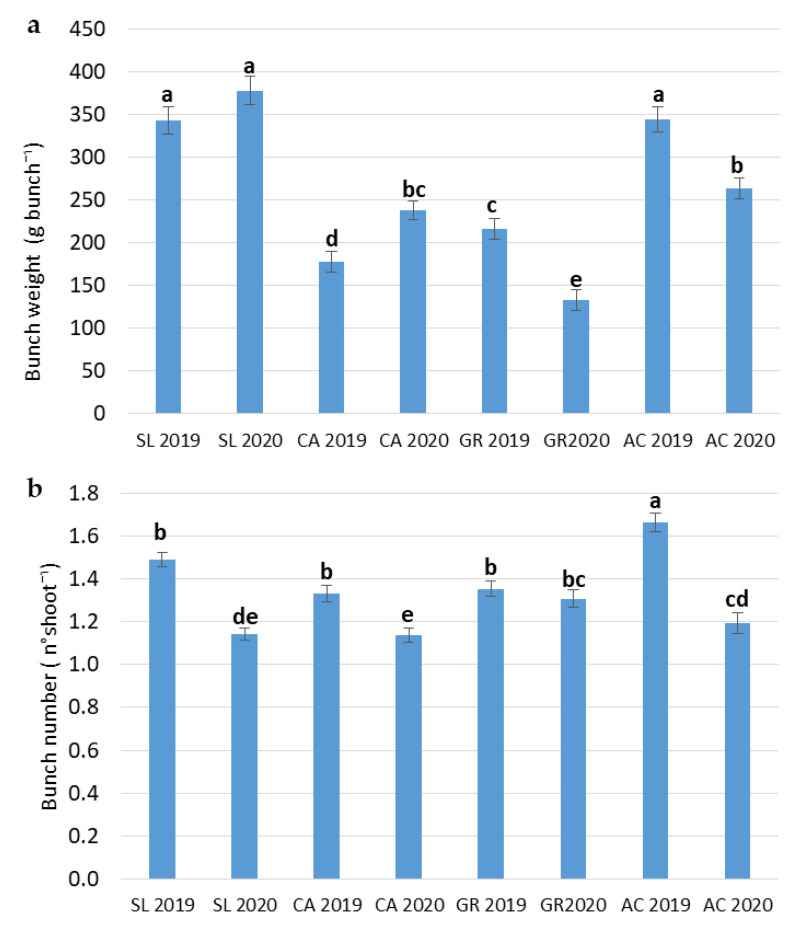
Combined effect of field and year (F x Y) on bunch weight (**a**) and bunch number (**b**) of *V. vinifera* subsp. *vinifera* ‘Falanghina’ vines at the four study sites: SL-Santa Lucia, CA-Calvese, GR-Grottole, AC-Acquafredde. Mean values and standard errors are shown. Different letters indicate significant differences according to Duncan’s multiple range test (*p* ≤ 0.05).

**Figure 2 plants-11-01507-f002:**
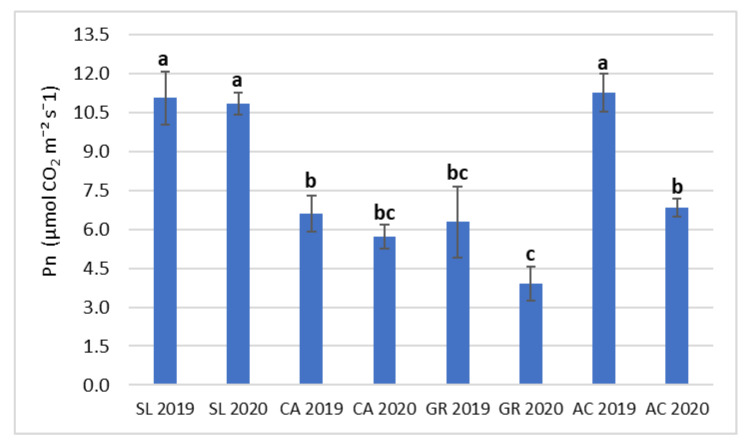
Combined effect of field and year (F x Y) on net photosynthetic rate (Pn) of V. vinifera subsp. vinifera ‘Falanghina’ vines at the four study sites: SL-Santa Lucia, CA-Calvese, GR-Grottole, AC-Acquafredde. Mean values and standard errors are shown. Different letters indicate significant differences according to Duncan’s multiple range test (*p* ≤ 0.05).

**Figure 3 plants-11-01507-f003:**
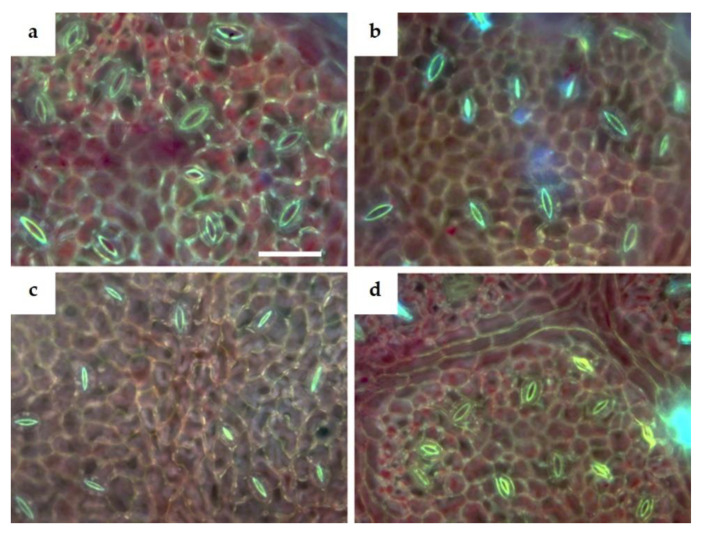
Epi-fluorescence microscopy views of abaxial leaf epidermis of V. vinifera ‘Falanghina’ vines at the four study sites: SL-Santa Lucia (**a**), CA-Calvese (**b**), GR-Grottole (**c**), AC-Acquafredde (**d**). Images are all at the same magnification. Bar = 50 µm.

**Figure 4 plants-11-01507-f004:**
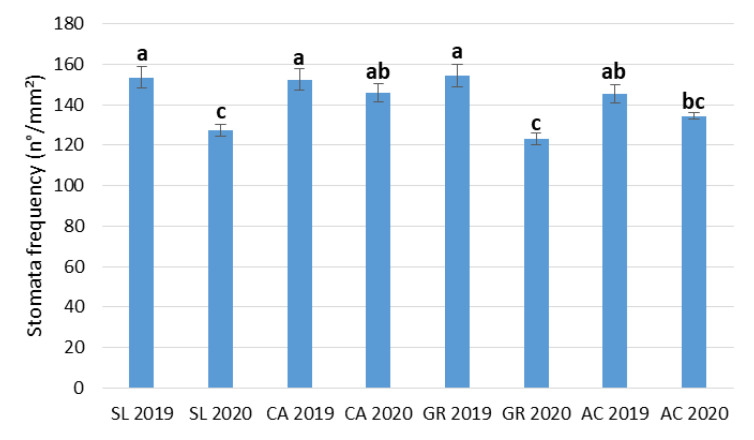
Combined effect of field and year (F x Y) on stomatal frequency of V. vinifera subsp. vinifera ‘Falanghina’ vines at the four study sites: SL-Santa Lucia, CA-Calvese, GR-Grottole, AC-Acquafredde. Mean values and standard errors are shown. Different letters indicate significant differences according to Duncan’s multiple range test (*p* ≤ 0.05).

**Figure 5 plants-11-01507-f005:**
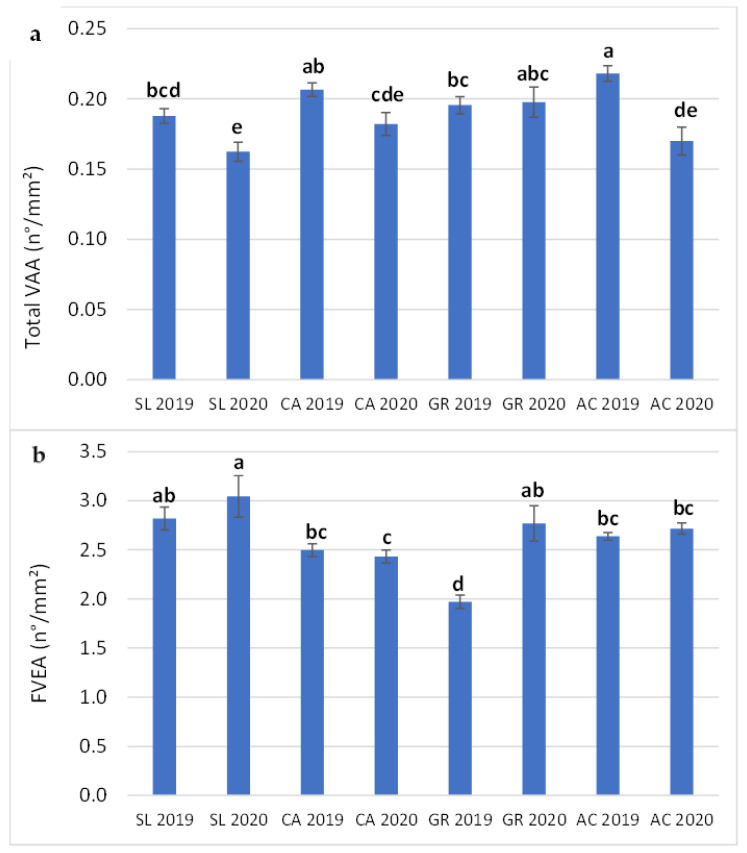
Combined effect of field and year (F x Y) on Total VAA (**a**) and FVEA (**b**) of V. vinifera subsp. vinifera ‘Falanghina’ vines at the four study sites: SL-Santa Lucia, CA-Calvese, GR-Grottole, AC-Acquafredde. Mean values and standard errors are shown. Different letters indicate significant differences according to Duncan’s multiple range test (*p* ≤ 0.05).

**Figure 6 plants-11-01507-f006:**
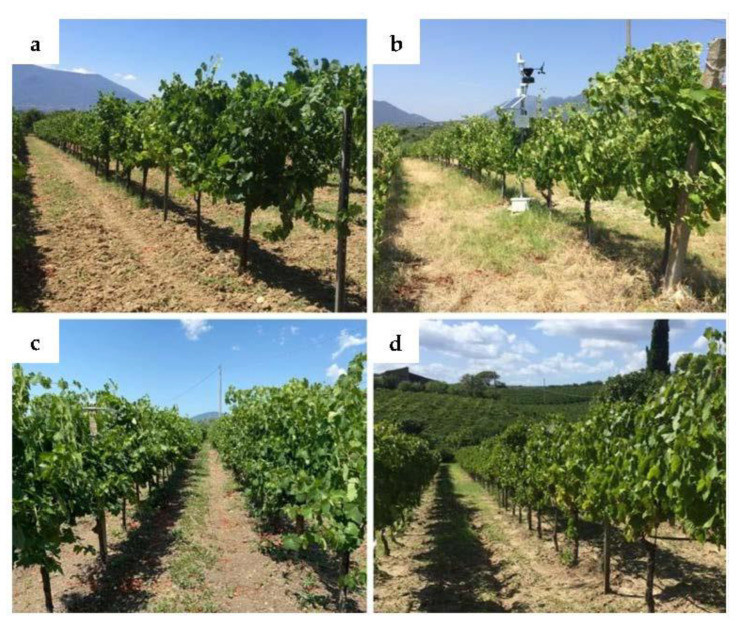
The four experimental sites Santa Lucia (**a**), Calvese (**b**), Grottole (**c**), Acquefredde (**d**) vineyards.

**Figure 7 plants-11-01507-f007:**
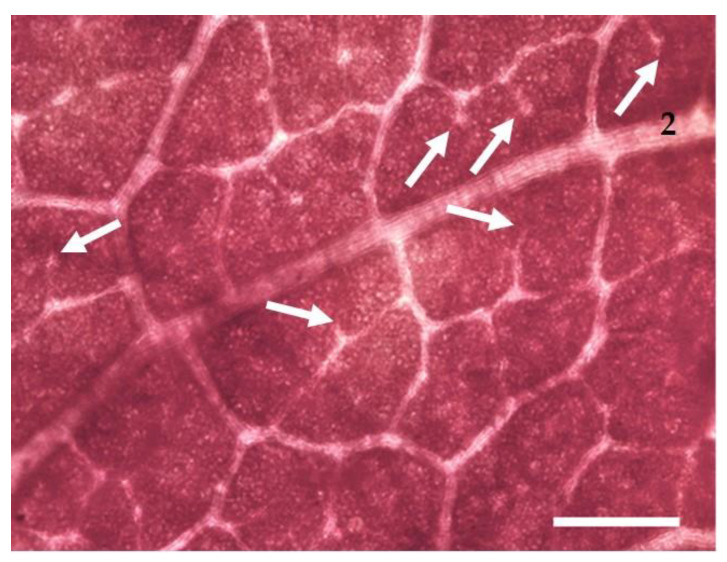
Light microscopy views of V. vinifera ‘Falanghina’ leaf lamina sample with arrows pointing to the FVEA (2, second-order vein). Bar = 300 µm.

**Table 1 plants-11-01507-t001:** Effects of field (F), year (Y), and their interaction (F x Y) on total shoot leaf area, single leaf area, shoot diameter, bunch weight, and bunch number per shoot of V. vinifera subsp. vinifera ‘Falanghina’ vines at the four study sites: SL-Santa Lucia, CA-Calvese, GR-Grottole, AC-Acquafredde. Mean values and standard errors are shown.

	Total Shoot Leaf Area	Single Leaf Area	Shoot Basal Diameter	Average Bunch Weight	Bunch Number
	(cm^2^ shoot^−1^)	(cm^2^)	(mm)	(g bunch^−1^)	(n° shoot^−1^)
Field (F)					
SL	4142 ± 250 ^a^	192 ± 10.4 ^a^	8.79 ± 0.16 ^ab^	360 ± 11.7 ^a^	1.3 ± 0.04 ^b^
CA	3416 ± 171 ^b^	135 ± 3.24 ^c^	7.89 ± 0.13 ^c^	208 ± 9.27 ^c^	1.2 ± 0.03 ^c^
GR	4194 ± 228 ^a^	151 ± 4.22 ^b^	9.02 ± 0.16 ^a^	175 ± 10.7 ^d^	1.3 ± 0.03 ^b^
AC	3320 ± 156 ^b^	156 ± 5.21 ^b^	8.49 ± 0.17 ^b^	304 ± 11.6 ^b^	1.4 ± 0.05 ^a^
Year (Y)					
2019	4308 ± 169 ^a^	168 ± 5.75 a	9.13 ± 0.10 ^a^	270 ± 10.8 ^a^	1.5 ± 0.02 ^a^
2020	3228 ± 107 ^b^	149 ± 3.42 b	7.97 ± 0.11 ^b^	253 ± 11.7 ^a^	1.2 ± 0.02 ^b^
Significance					
Field (F)	***	***	***	***	***
Year (Y)	***	**	***	NS	***
F x Y	NS	NS	NS	***	***

NS, **, and ***, Not significant or significant at *p* < 0.01, and 0.001, respectively. Different letters within each column indicate significant differences according to Duncan's multiple comparison tests (*p* ≤ 0.05).

**Table 2 plants-11-01507-t002:** Effects of field (F), year (Y) and their interaction (F x Y) on net photosynthetic rate (Pn), stomatal conductance (gs), substomatal CO_2_ concentration (Ci), leaf transpiration rate (E), instantaneous water use efficiency (_in_WUE), electron transport rate (ETR), quantum yield of PSII linear electron transport (ΦPSII), and maximum quantum efficiency of PSII photochemistry (Fv/Fm) of V. vinifera subsp. vinifera ‘Falanghina’ vines at the four study sites: SL-Santa Lucia, CA-Calvese, GR-Grottole, AC-Acquafredde. Mean values and standard errors are shown.

	Pn	gs	Ci	E	inWUE	ETR	ΦPSII	Fv/Fm
	(µmol CO_2_ m^−2^s^−1^)	(mmol H_2_O m^−2^ s^−1^)	(µmol CO_2_ mol^−1^)	(mol H_2_O m^−2^ s^−1^)	(µmol CO_2_/ mol H_2_O)			
Field (F)								
SL	10.9 ± 0.66 ^a^	183.5 ± 19.1 ^a^	267.6 ± 12.5 ^a^	5.00 ± 0.31 ^a^	2.55 ± 0.26 ^a^	172.1 ± 5.30 ^a^	0.309 ± 0.007 ^a^	0.784 ± 0.003 ^a^
CA	6.20 ± 0.49 ^c^	103.6 ± 12.1 ^b^	228.3 ± 16.0 ^a^	3.37 ± 0.34 ^b^	1.97 ± 0.20 ^a^	137.3 ± 3.40 ^b^	0.250 ± 0.006 ^b^	0.752 ± 0.003 ^b^
GR	5.15 ± 0.78 ^c^	66.7 ± 10.9 ^b^	237.9 ± 17.8 ^a^	3.33 ± 0.37 ^b^	1.83 ± 0.32 ^a^	135.1 ± 6.19 ^b^	0.242 ± 0.010 ^b^	0.754 ± 0.004 ^b^
AC	9.31 ± 0.61 ^b^	159.4 ± 15.5 ^a^	225.2 ± 15.3 ^a^	5.31 ± 0.24 ^a^	1.98 ± 0.17 ^a^	181.2 ± 4.81 ^a^	0.325 ± 0.008 ^a^	0.788 ± 0.003 ^a^
Year (Y)								
2019	8.77 ± 0.58 ^a^	131.5 ± 14.0 ^a^	214.6 ± 13.6 ^b^	4.60 ± 0.27 ^a^	2.26 ± 0.21 ^a^	153.8 ± 5.14 ^a^	0.279 ± 0.008 ^a^	0.772 ± 0.004 ^a^
2020	6.78 ± 0.45 ^b^	120.4 ± 8.8 ^a^	267.4 ± 4.10 ^a^	4.00 ± 0.22 ^a^	1.74 ± 0.09 ^b^	159.3 ± 3.98 ^a^	0.284 ± 0.008 ^a^	0.766 ± 0.002 ^a^
Significance								
Field (F)	***	***	NS	***	NS	***	***	***
Year (Y)	**	NS	***	NS	*	NS	NS	NS
F x Y	*	NS	NS	NS	NS	NS	NS	NS

NS, *, **, and ***, Not significant or significant at *p* < 0.05, 0.01, and 0.001, respectively. Different letters within each column indicate significant differences according to Duncan’s multiple comparison tests (*p* ≤ 0.05).

**Table 3 plants-11-01507-t003:** Effects of field (F), year (Y), and their interaction (F x Y) on stomata traits of V. vinifera subsp. vinifera ‘Falanghina’ vines at the four study sites: SL-Santa Lucia, CA-Calvese, GR-Grottole, AC-Acquafredde. Mean values and standard errors are shown.

	Stomata Length	Stomata Width	Stomata Frequency
	(µm)	(µm)	(n/mm^2^)
Field (F)			
SL	33.2 ± 0.43 ^a^	19.2 ± 0.31 ^a^	140.3 ± 3.74 ^a^
CA	29.6 ± 0.34 ^b^	16.9 ± 0.23 ^b^	149.2 ± 3.54 ^a^
GR	27.2 ± 0.55 ^c^	15.5 ± 0.36 ^c^	138.6 ± 4.04 ^a^
AC	24.8 ± 0.38 ^d^	14.3 ± 0.21 ^d^	139.9 ± 2.50 ^a^
Year (Y)			
2019	28.8 ± 0.37 ^a^	16.1 ± 0.22 ^b^	151.4 ± 2.56 ^a^
2020	28.6 ± 0.36 ^a^	16.8 ± 0.24 ^a^	132.6 ± 1.88 ^b^
Significance			
Field (F)	***	***	NS
Year (Y)	NS	*	***
F x Y	NS	NS	*

NS, * and ***, Not significant or significant at *p* < 0.05 and 0.001, respectively. Different letters within each column indicate significant differences according to Duncan’s multiple comparison tests (*p* ≤ 0.05).

**Table 4 plants-11-01507-t004:** Effects of field (F), year (Y), and their interaction (F x Y) on vein traits in leaves of V. vinifera ‘Falanghina’ vines at the four study sites: SL-Santa Lucia, CA-Calvese, GR-Grottole, AC-Acquafredde. Mean values and standard errors are shown.

	Minor VLA	Major VLA	Total VLA	Minor VAA	Major VAA	Total VAA	FVEA
	(mm/mm^2^)	(mm/mm^2^)	(mm/mm^2^)	(mm^2^/mm^2^)	(mm^2^/mm^2^)	(mm^2^/mm^2^)	(n/mm^2^)
Field (F)							
SL	2.31 ± 0.06 ^a^	0.729 ± 0.036 ^a^	2.89 ± 0.06 ^c^	0.118 ± 0.004 ^c^	0.065 ± 0.004 ^a^	0.175 ± 0.005 ^b^	2.93 ± 0.11 ^a^
CA	2.59 ± 0.01 ^a^	0.865 ± 0.049 ^a^	3.27 ± 0.08 ^a^	0.132 ± 0.004 ^b^	0.072 ± 0.004 ^a^	0.194 ± 0.005 ^a^	2.47 ± 0.05 ^c^
GR	2.47 ± 0.09 ^a^	0.751 ± 0.054 ^a^	3.04 ± 0.07 ^bc^	0.134 ± 0.006 ^b^	0.072 ± 0.004 ^a^	0.197 ± 0.005 ^a^	2.37 ± 0.10 ^c^
AC	2.58 ± 0.14 ^a^	0.817 ± 0.042 ^a^	3.22 ± 0.12 ^ab^	0.145 ± 0.004 ^a^	0.069 ± 0.003 ^a^	0.194 ± 0.007 ^a^	2.68 ± 0.03 ^b^
Year (Y)							
2019	2.45 ± 0.06 ^a^	0.799 ± 0.028 ^a^	3.07 ± 0.05 ^a^	0.137 ± 0.003 ^a^	0.075 ± 0.002 ^a^	0.202 ± 0.003 ^a^	2.48 ± 0.04 ^b^
2020	2.52 ± 0.09 ^a^	0.781 ± 0.039 ^a^	3.14 ± 0.07 ^a^	0.127 ± 0.004 ^b^	0.064 ± 0.003 ^b^	0.178 ± 0.004 ^b^	2.74 ± 0.06 ^a^
Significance							
Field (F)	NS	NS	*	**	NS	**	***
Year (Y)	NS	NS	NS	*	*	***	**
F x Y	NS	NS	NS	NS	NS	**	**

NS, *, **, and ***, Not significant or significant at *p* < 0.05, 0.01, and 0.001, respectively. Different letters within each column indicate significant differences according to Duncan’s multiple comparison tests (*p* ≤ 0.05).

## Data Availability

The data supporting the findings of this study are available from the corresponding authors (V.D.M. and C.C.) upon reasonable request.

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
