# Peer review of "How Leaf Vein and Stomata Traits Are Related with Photosynthetic Efficiency in Falanghina Grapevine in Different Pedoclimatic Conditions"

_plants, 2022, doi:10.3390/plants11111507_

Round 1
Reviewer 1 Report
My comments can be found in the attached PDF.

Author Response
Q1. Novelty. The Authors mostly fail in conveying the novelty of this work. A Conclusion section would probably help in summarizing the main contribution.
R1: We thank the reviewer #1 for suggesting to better highlight the novelty of the content of the paper. To address this important point we added a conclusion section to summarize the contribution of the work to the state of art recalling the point of novelty reported in the end of Introduction section. Lines 549-560.
Q2. Role of other morphological adaptations. The role (or possible role) of other morphological adaptations to different pedoclimatic conditions should be at least mentioned. Plants growing under water stress, for example, tend to invest in the belowground biomass at the expense of canopy growth (Friedlingstein et al. 1999, Bacelar, Moutinho-Pereira et al. 2007).
R2. We added some considerations and references. Lines 50-54.
Q3. Role of other sources of stress. The role of other sources of abiotic stress rather than water stress is not even mentioned. For example, there is growing evidence and concern about the importance of salt stress (Perri, Suweis et al. 2018, Hassani, Azapagic et al. 2020, Perri, Suweis et al. 2020), which could substantially affect plants’ gas exchange and lead to leaf anatomy adjustment similar to those observed in response to water stress (Huang, Li et al. 2019, Perri, Katul et al. 2019). I believe this should be mentioned.
R3: We added some considerations and references. Lines 54-57.
Q4. Water use efficiency. I wonder why WUE was not calculated and shown since measurements of both water and CO2 fluxes are available. The implications for WUE remain vague in the manuscript.
R4 : We calculated instantaneous WUE and added data in table 3. We added related sentences in materials and methods, results and discussion. Lines 183-187, 375-379, 467-468, Table 3.
Q5. Results presentation and Figures. The presentation of the results and the Figures are poor. Many Figures basically duplicate the information already reported in tables. I do not think both are needed. I would suggest using only tables when the graphics don does not improve the visualization of the results (Figures 2, 4, and 5).
R5 : We thank the reviewer for suggesting to better arrange figures and tables, but we would like to drive his/her attention on the fact that the data reported in the figures are not a duplicate of the ones in the tables. Indeed, tables showed only the averages effects of main factors (sites and years), while in the figures the effects of significant interactions between the two main factors are reported (results of one-way-anova: please fer to what is reported in Materials and Methods at lines 543-544). Therefore, we think this is an important information and we do prefer to keep the figures.
Q6:I believe that if these comments are addressed, the manuscript could represent a valuable contribution.
R6: We thank the reviewer for the valuable comments which helped us to improve the manuscript.
Reviewer 2 Report
The present study highlight the relationship between leaf vein as well stomata traits and photosynthetic efficiency in cv. Falanghina in contrasting pedoclimatic conditions.
The topic is relevant and of great interest within the scientific community and Plants readers. The results and consistent but more attention could have been posed on the following points.
- While a weather station and soil water content data are reported. No statistical analysis was performed among sites to really assess if the supposed or expected stress level was significantly different among sites within each season. The study would have benefit of including leaf water potential measurements at key stages (instead of bunch weight, as an example).
- While vegetative parameters measured at véraison (shoot length, leaf area, shoot caliper) are consistent and could be considered as proxy of the physiological conditions experienced by the vines, at this phenological phase (it would have been important to include the doy of measurements and even to monitor phenology and report it on a BBCH scale), to measure bunch weight at this stage it does not make sense. I would have, on the contrary, collect it (on the same shoots) at harvest. I would remove it.
- The biometrical and ecophysiological (gas exchange in particular) methods part is poor, plenty of errors and must be rewritten (see minor comments later) including important missing operative details:
- air VPD in the leaf chamber
- flow rate
- chamber air temperature
- Does any canopy management was performed in the vineyard (e.g. shoot trimming)?
For the above comments and the minor detailed ones below I suggest at least Major Revisions to be addressed before considering the study as suitable to be published in Plants.
Minor detailed comments:
- L139: total 'shoot' leaf area; (please refer in the text as it and not as total leaf area)
- L138: please report sites row orientation that could have affect radiation interception regimes
- L153: yield? at veraison?
- L172: 'but' ??????
- L175: change 'plants' in 'leaves'
- Table3: include CO2 in the unit for Pn and Ci, and H2O for gs and E
- Figure 2: add CO2, check the remaining figures as well
- L380: are/were placed
- L424: I would not use "the canopy was estimated" but "characterized"
- L425: I would suggest "on two shoots" since it is implied that on a double guyot pruning system, both a annual fruit canes
- L425-426: remove 'holding the production of the year'. Why was averaged on all bunches of the vineyard and not on the selected 20 vines?
- L428: an allometric equation based only on 80 leaves is quite poor. Normally hundreds of leaves are necessary to establish a consistent relationship. A figure reporting measured vs estimated measurements and the related RMSE must be reported as supplementary material.
- L436: remove 'one'
- L435-438: Insert somewhere 'in each site'
- L439 'but' ?????again?
- L440: room temperature??????
- L442: '2' apice
- L435-442: rewrite entirely
- L445: 'was' performed
- L446: 'weak'??? or 'dark'??? it is quite different
Author Response
We are grateful to the reviewer for his/her comments which helped us to improve the manuscript. We hope the new version will fulfill his/her expectations.
- Q1 While a weather station and soil water content data are reported. No statistical analysis was performed among sites to really assess if the supposed or expected stress level was significantly different among sites within each season. The study would have benefit of including leaf water potential measurements at key stages (instead of bunch weight, as an example).
- R1: We installed one weather station per each site and therefore we do not have replicates to perform statistical analysis. The station was only for monitoring purposes to describe the sites. About the leaf water potential measurements, we agree the study would have benefited from its measurement but, unfortunately, we did not have the possibility to measure it. However, the analysis of d13C in the same 4 vineyards in a previous study showed that the vines at the different sites were characterized by different levels of drought stress (please refer to reference n. 33). We will keep in mind the reviewer suggestion for future work.
- Q2 While vegetative parameters measured at véraison (shoot length, leaf area, shoot caliper) are consistent and could be considered as proxy of the physiological conditions experienced by the vines, at this phenological phase (it would have been important to include the doy of measurements and even to monitor phenology and report it on a BBCH scale), to measure bunch weight at this stage it does not make sense. I would have, on the contrary, collect it (on the same shoots) at harvest. I would remove it.
R2. We are really very sorry for this misunderstanding. Of course we measured bunch weight at harvest but we were not clear in the methods. We clarified this in Materials and Methods. Lines 449-450.
- Q3 The biometrical and ecophysiological (gas exchange in particular) methods part is poor, plenty of errors and must be rewritten (see minor comments later) including important missing operative details:
- air VPD in the leaf chamber
- flow rate
- chamber air temperature
R 3: We carefully checked this section and added requested information. Lines 457-470.
- Q4 Does any canopy management was performed in the vineyard (e.g. shoot trimming)?
R4 Line 397-398. Shoot trimming was performed once after fruit set. Lines 407-408.
For the above comments and the minor detailed ones below I suggest at least Major Revisions to be addressed before considering the study as suitable to be published in Plants.
Minor detailed comments:
- Q5 L139: total 'shoot' leaf area; (please refer in the text as it and not as total leaf area)
- R5 Done
- Q6. L138: please report sites row orientation that could have affect radiation interception regimes
- R6. Done. Lines 408-409.
- Q7. L153: yield? at veraison?
- R7. Please refer to R2.
- Q8. L172: 'but' ??????
- R8. This form is used to indicate “with the exception of”. However, we rephrased.
- Q9. L175: change 'plants' in 'leaves'
- R9. Done
- Q10 Table3: include CO2 in the unit for Pn and Ci, and H2O for gs and E
- R10. Done
- Q11. Figure 2: add CO2, check the remaining figures as well
- R11. Done.
- Q12. L387: are/were placed
- R12. Done
- Q13. L424: I would not use "the canopy was estimated" but "characterized"
- R13. Done.
- Q14 L425: I would suggest "on two shoots" since it is implied that on a double guyot pruning system, both a annual fruit canes
- R14 Done.
- Q15 L425-426: remove 'holding the production of the year'. Why was averaged on all bunches of the vineyard and not on the selected 20 vines?
- R15. The parameters were calculated on the 20 plants. We removed misleading parts.
- Q16. L428: an allometric equation based only on 80 leaves is quite poor. Normally hundreds of leaves are necessary to establish a consistent relationship. A figure reporting measured vs estimated measurements and the related RMSE must be reported as supplementary material.
- R16. We applied an allometric model already established for grapevine using a limited number of leaves to avoid removing too many leaves. We considered a number of leaves which gave us a robust relationship as indicated by R2 values. Since this method is commonly used in grapevine (please see references reported in Materials and Methods, line 454), we would avoid reporting the regression lines and statistics in the supplementary material but we attach graphs to this revision for your perusal (file: attachment-to-R16.doc).
- Q17 L436: remove 'one'
- R17 Done
- Q18. L435-438: Insert somewhere 'in each site'
- R18. Done
- Q19 L439 'but' ?????again?
- R19 Deleted
- Q20 L440: room temperature??????
- R20. Of course it was not room temperature, sorry for this. We amended the text.
- Q21. L442: '2' apice
- R21. Done
- Q22. L435-442: rewrite entirely
- R22. We rewrote the whole paragraph. Lines 457-470.
- Q23. L445: 'was' performed
- R23. Done.
- Q24 L446: 'weak'??? or 'dark'??? it is quite different
- R24. “weak” is correct. The weak measuring light of 3, 4 µmol photons m2 s¯¹ was used to induce the ground fluorescence signal, F0, on dark adapted leaves, as specified in the text.

Round 2
Reviewer 2 Report
The revised manuscript improved and resulted now more focused and significant. Before considering it as suitable to be published in Plants, I recommend:
- to include accurate phenological stages (BBCH) at which measurements and samplings were performed.
- throughout the manuscript, including Figures, Tables, the authors should adhere to significant digits (decimals). All values should be reported to the scale that the data is known to, computer programs or weather stations will likely output data into many more decimals than is appropriate. Please homogenize the use of significant digits (decimal) in the manuscript and tables (i.e. reporting a bunch number of 1.318 it does not make sense, Table 2).
Author Response
Dear Reviewer,
thanks for your fruitful comments. We have followed your advice by inserting the BBCH reference in the Materials and Methods to better identify phenological phases and adjusted decimals throughout the manuscript.
New modifications are highlighted in blue.
Best regards
--------------
Detailed responses:
Reviewer n. 2
C1. The revised manuscript improved and resulted now more focused and significant.
R1. We thank the Reviewer 2 for his/her comments. The fruitful criticisms helped us to improve the manuscript.
C2. Before considering it as suitable to be published in Plants, I recommend: to include accurate phenological stages (BBCH) at which measurements and samplings were performed.
R2. We added requested information in Materials and methods
C3. throughout the manuscript, including Figures, Tables, the authors should adhere to significant digits (decimals). All values should be reported to the scale that the data is known to, computer programs or weather stations will likely output data into many more decimals than is appropriate. Please homogenize
the use of significant digits (decimal) in the manuscript and tables (i.e. reporting a bunch number of 1.318 it does not make sense, Table 2).
R3. We revised all tables and figures homogenizing decimals